# The Design of a 1–6 GHz Broadband Double-Balanced Mixer

**DOI:** 10.3390/mi14112069

**Published:** 2023-11-07

**Authors:** Yujun Wang, Jianglai Tian, Bo Wang, Lixi Wan, Zhi Jin

**Affiliations:** 1Institute of Microelectronics of Chinese Academy of Sciences, Beijing 100029, China; wangyujun@ime.ac.cn; 2University of Chinese Academy of Sciences, Beijing 100029, China; 3Chengdu Tiger Microelectronics Research Institute Co., Ltd., Chengdu 610000, China

**Keywords:** Marchand balun, impedance, broadband, mixer

## Abstract

This brief proposes a 1–6 GHz broadband double-balanced mixer. On the basis of the standard Marchand balun mixer, two techniques to enhance the performance of the mixer are proposed. Firstly, by loading capacitors, the amplitude and phase imbalance of the balun at high frequencies are improved, thereby expanding the relative bandwidth of the mixer by 0.1. Secondly, by cascading a first-order LC filter at the intermediate-frequency (IF) port, the leakage of radio frequency (RF) and local oscillator (LO) signals at the IF port is reduced by approximately 8 dB and 7 dB, respectively. This brief analyzes the various parameters that affect balun indicators and designs a broadband balun with a double-layer spiral line structure. As a result of these technologies, a 1–6 GHz double-balanced mixer is realized. The chip area is 1.13 mm × 0.95 mm, and the isolation between the IF port and the RF port is 34 dB and yields 13.5 dBm input power at 1 dB compression point (IP1 dB). The chip is fabricated via the GaAs pHEMT process.

## 1. Introduction

A passive double-balanced mixer does not require a DC power supply and, therefore, does not introduce noise, making it a commonly used mixing structure in communication systems. However, in communication systems such as missile-borne data links and unmanned aerial vehicles, mixers require high bandwidth and miniaturization requirements, which are difficult to meet simultaneously. The main constraint is the design of the balun, and the isolation and linearity of mixers are also particularly important for communication systems. Therefore, many scholars have conducted research on this issue.

In [1], a 0.7–7.7 GHz mixer was designed using a microstrip line and wide-edge coupled suspension line balun structure, but the balun size of the microstrip line structure was relatively large, with a mixer size of 5.0 cm × 3.8 cm. In [2], a dual Marchand balun mixing ring structure mixer with a frequency range of 1.5–5.0 GHz was designed, and a filter compensation network was added. Due to the use of a double mixing ring structure, the size was still large at 1.6 mm × 1.6 mm. In [3], the object of study was a variant of the standard Marchand balun structure, which uses a 1/4 coupling line on the RF side and adds a set of parallel inductance and resistance compensation structures to achieve a 2–22 GHz ultra-wideband mixer. However, this structure cannot simultaneously meet the isolation of the IF port to the RF port and the IF port to the LO port. In [4], a 2.8–6 GHz mixer was designed using an asymmetric helical Marchand balun structure, achieving good isolation while maintaining a smaller size. However, the relative bandwidth of the mixer was relatively low. In [5], a 27–44 GHz double-balanced mixer was designed using a microstrip line Marchand balun. In order to obtain more accurate balun characteristics, a full-wave electromagnetic simulator was used in HFSS to simulate the balun. In [6], a 26–55 GHz broadband MMIC compatible uniplanar balun was designed using a biased air gap coupler. It achieved insertion loss below 2 dB, with an amplitude and phase imbalance less than ±1 dB and 5°. In order to improve the isolation between the IF port and the RF port, as well as between the IF port and the LO port, a second-order LC low-pass filter was added to the IF port in [7], but the filter area accounted for about one-third of the entire chip.

Due to the fact that the electrical length of each line in a Marchand balun is theoretically a quarter wavelength, the standard Marchand balun bandwidth is limited. When it deviates from the required quarter wavelength, the amplitude and phase balance will decrease. This brief studies balun theory and analyzes the effects of coupling line width and line spacing on the odd mode impedance and coupling factor. By loading capacitors at the balun port, we make the impedance of the balun better matched to the impedance of the quad-ring diode at high frequencies, and the amplitude and phase imbalance in the high-frequency band are improved, thereby expanding the relative bandwidth by 0.1; in addition, without increasing the chip size, by cascading a first-order LC filter at the IF output port, the isolation problem at the tap of the double-balanced mixer is improved, resulting in an 8 dB reduction in the leakage intensity of the RF signal and a 7 dB reduction in the leakage intensity of the LO signal.

Finally, with a size of 1.13 mm × 0.95 mm, a 1–6 GHz broadband frequency mixer was achieved with an isolation of approximately 34 dB from the RF and LO ports to the IF port, and an IP1 dB of 13.5 dBm. To our knowledge, these data represent the optimal combination of bandwidth, isolation, linearity, and size for passive broadband double-balanced mixers.

The brief is organized as follows: Section 2 provides a detailed introduction to the conditions for an ideal balun and the working principle of the mixer. Section 3 introduces the design of the balun and the design and optimization of the mixer. Section 4 compares the simulation results and measurement results of the mixer. Section 5 is a brief introduction to the conclusion.

## 2. Principle Design

### 2.1. Typical Marchand Balun Principle Circuit

Figure 1 shows the structure of a typical Marchand balun. It consists of two coupled lines of 1/4λ length; P1 is an unbalanced input terminal, and P2 and P3 are balanced output terminals. One end of the two coupled lines is grounded, and one end is suspended or grounded (for the convenience of expression, it is referred to as the open port in this brief).

As shown in Figure 1, the balun is a three-port passive device. When analyzing its network parameters, it can be analyzed as a four-port network, and one of the ports can be set to open or short. Shown in Figure 2 is a schematic diagram of a four-port network; port 3 is short-circuited or open-circuited, port 1 is an unbalanced input port, and ports 2 and 4 are balanced output ports. 

The network shown in Figure 2 can be analyzed using the odd–even mode method [8], as shown in Figure 3 and Figure 4. 

*T* is the transmission coefficient, and Γ is the reflection coefficient.

Ideally, the input port is matched, and the power of the two differential output ports is equal, which is half of the input port, and the *S* parameter is expressed as: $S_{11}=0$, $S_{21}={-S}_{31}$ [9]; using balun’s *S* parameter matrix theory [10], it can be deduced and expressed as Equations (1) and (2):(1)$$\frac{\Gamma_{e1}+\Gamma_{o1}-2\Gamma \Gamma_{e1}\Gamma_{o1}}{2-\Gamma \left(\Gamma_{e1}+\Gamma_{o1}\right)}=0$$
(2)$$\frac{2T_{e}\left(1-\Gamma \Gamma_{o1}\right)}{2-\Gamma \left(\Gamma_{e1}+\Gamma_{o1}\right)}=0$$

In Equation (1), assuming that the third port is opened or grounded, that is, the third port reflection coefficient Γ = ±1, Equation (1) is simplified as:(3)$$\Gamma_{e1}+\Gamma_{o1}\pm 2\Gamma_{e1}\Gamma_{o1}=0$$

According to the definition of the reflection coefficient,
$$\Gamma_{e1}=\frac{Z_{0e}-Z_{in}}{Z_{0e}+Z_{in}}$$
$$\Gamma_{o1}=\frac{Z_{0o}-Z_{in}}{Z_{0o}+Z_{in}}$$

Substituting this into Equation (3), we obtain: (4)$$Z_{0e}+Z_{0o}=2\times Z_{in}\;(\mathrm{while}\;\Gamma =+1)\;$$
(5)$$Z_{in}=\frac{2Z_{0e}Z_{0o}}{Z_{0e}+Z_{0o}}\;(\mathrm{while}\;\Gamma =-1)$$

It can be found that, when the ports are in an ideal matching situation, for a 50-ohm system, Equations (4) and (5) can be expressed as:(6)$$Z_{0e}+Z_{0o}=100\;(\mathrm{while}\;\Gamma =+1)\;$$
(7)$$Z_{0e}=-\frac{25Z_{0o}}{25-Z_{0o}}\;(\mathrm{while}\;\Gamma =-1)$$

Meanwhile, Equation (2) shows that $T_{e}=0$; that is, the transmission coefficient of the even mode is 0, and the even mode impedance is infinite.

Therefore, when designing a balun, it is necessary to increase the even-mode impedance as much as possible to suppress the transmission of common-mode signals. It can be concluded from Equation (7) that if port 3 is short-circuited, a larger even-mode impedance can be obtained when the odd-mode impedance is close to 25 ohms, which can make the differential-mode signal transmit smoothly at the same time.

Using the coupling factor to express the *S* parameter network matrix of the balun, under ideal conditions, the *S* matrix of the Marchand balun can be expressed as follows [11]:(8)$${\left[S\right]}_{balun}=\left[\begin{array}{ccc} \frac{1-3C^{2}}{1+C^{2}} & j\frac{2C\sqrt{1-C^{2}}}{1+C^{2}} & -j\frac{2C\sqrt{1-C^{2}}}{1+C^{2}}\\ j\frac{2C\sqrt{1-C^{2}}}{1+C^{2}} & \frac{1-C^{2}}{1+C^{2}} & \frac{2C^{2}}{1+C^{2}}\\ -j\frac{2C\sqrt{1-C^{2}}}{1+C^{2}} & \frac{2C^{2}}{1+C^{2}} & \frac{1-C^{2}}{1+C^{2}} \end{array}\right]$$

Here, ‘*C*’ is the coupling factor; it can be seen from Equation (8) that, in order to make port 1 matched, ‘*C*’ needs to satisfy:(9)$$\frac{1-3C^{2}}{1+C^{2}}=0$$

At this time, ‘*C*’ is equal to $1/\sqrt{3}$, because port 1 cannot match it exactly, so ‘*C*’ should be increased as much as possible to make ‘*C*’ close to $1/\sqrt{3}$ in order to achieve good matching and lower conversion loss.

During MMIC (Monolithic Microwave Integrated Circuit) design, in view of the large size of the microstrip line, the balun can be generated through a spiral line. Due to the existence of mutual inductance and mutual capacitance in the spiral line method, for the same line length, the odd-mode impedance will be increased, and the coupling factor will be reduced. It can be adjusted by changing the coupling line width and coupling line spacing. Figure 5 depicts the effects of coupling line width and coupling line spacing on the odd-mode impedance and coupling factor. Considering that the minimum line spacing in this manufacturing process can reach 3.5 μm, a certain margin needs to be reserved, and the minimum spacing here was set to 4 μm. Fortunately, simply because of the mutual capacitance and inductance, the resonant frequency of the line per unit length is lower, which brings the advantage that at a given frequency, the size of the spiral balun can be smaller. 

### 2.2. Principle of Double-Balanced Mixer

As shown in Figure 6, two baluns complete the conversion of RF/LO from single-ended to differential (unbalanced to balanced), four identical diodes are connected in series to form a loop, and the IF is led out from the center tap point of any baluns, generally from the RF-side balun, in order to improve the LO-IF isolation. The midpoint of the LO-side balun is grounded. It is also possible to directly lead out the center points of the two baluns; at this time, the IF is the balanced output.

Figure 6 shows the loading of two 1.38 pF capacitors and one 0.8 nH inductor on the basis of a typical Marchand balun structure mixer to expand the frequency band and improve isolation, which will be discussed in the following chapters.

The diode plays the role of frequency mixing in the mixer. The following is a brief analysis of its working principle. Figure 7 is a schematic diagram of the principle of diode mixing. In the figure, the input voltages ${2v}_{S}$ and ${2v}_{L}$ are, respectively, input at the RF and LO ports.

Suppose $v_{S}=V_{S}cos\left(\omega_{S}t\right)$, $v_{L}=V_{L}cos\left(\omega_{L}t\right)$, $v_{S}\ll v_{L}$, and the characteristic of the diode is $i=f\left(v\right)$; then, the diode current is:(10)$$i=f\left(v\right)=f\left(v_{S}+v_{L}\right)$$

Expand Equation (10) according to the Taylor series to obtain:(11)$$i=f\left(v_{L}\right)+f^{\left(1\right)}\left(v_{L}\right)v_{S}+\frac{1}{2!}f^{\left(2\right)}\left(v_{L}\right)v_{S}^{2}+\cdots +\frac{1}{m!}f^{\left(m\right)}\left(v_{L}\right)v_{S}^{m}+\cdots$$

In Equation (11), $f\left(v_{L}\right)$ is the diode current when only the local oscillator voltage is applied,
(12)$$f^{\left(m\right)}\left(v_{L}\right)=\sum_{n=-\infty }^{\infty }y_{m,n}e^{jn\omega_{L}t}$$
(13)$$v_{S}=V_{S}cos\left(\omega_{S}t\right)=\frac{1}{2}V_{S}\left[e^{j\omega_{S}t}+e^{-j\omega_{S}t}\right]$$

Substituting Equations (12) and (13) into Equation (11) and completing the calculation, the IF port current can be obtained as follows:(14)$$i_{0}\left(t\right)=\sum_{n=-\infty }^{\infty }\sum_{m=-\infty }^{\infty }{\dot{I}}_{n,m}\exp\left[j\left(n\omega_{L}+m\omega_{S}\right)t\right]\cdot \left(1-e^{jn\pi }\right)\left(1-e^{jm\pi }\right)$$

It can be seen from Equation (14) that when ‘*n*’ and ‘*m*’ are even numbers, then $i_{0}\left(t\right)=0$; that is to say, under this structure, the combined component of the even harmonics obtained by mixing the LO and RF is completely suppressed, and only the combined components of the odd harmonics are left. The advantage of this structure is that the IF spurs are small.

## 3. Circuit Design

### 3.1. Designing Process

#### 3.1.1. The Design of the Balun

Generally speaking, the RF balun and LO balun are symmetrical and interchangeable. They are designed using the method of double-wire winding, and Figure 8 shows a schematic diagram of double-wire winding.

Firstly, we designed the initial value of the length and width of the balun according to the operating frequency. The total length of the balun was designed to be a quarter of the working wavelength. The total length of the balun can be expressed as Equation (15): (15)$$L_{balun}=2n\left(a+b\right)+8n(w+s)$$

Here ‘*n*’ represents the number of balun turns, ‘*a*’ is the unilateral length of the balun inner ring, ‘*b*’ is the unilateral width of the balun inner ring, ‘*w*’ is the balun line width, and ‘*s*’ is the balun line distance, which can be designed to be as small as possible to increase the even-mode impedance and reduce the odd-mode impedance.

According to the analysis in Section 2.1, the parameters of the balun need to be adjusted to make the odd-mode impedance closer to 25 ohms, and the two balanced ports almost have the same amplitude and 180° phase difference in the working-frequency band.

The design frequency of the balun is 1–6 GHz, using two layers of metal interwinding to greatly reduce the odd-mode impedance. At the same time, under the limitation of chip-processing technology, the distance between the lines should be minimized as much as possible to further reduce the odd-mode impedance, ultimately making it close to 25 ohms in the operating-frequency band. A comparison of the odd-mode impedance curves under different conditions is shown in Figure 9.

From the odd-mode impedance curve, it can be seen that the curve is in the shape of a bathtub. The more it deviates from a quarter of the wavelength, the more the odd-mode impedance deviates from 25 ohms. Affected by the mutual capacitance between the spiral lines at the high end, the high-end deviation is small, and the lower the end, the greater the impedance deviation. 

The final coupling line pattern is shown in Figure 10a; using double-layer coupling, the line width of the TOP layer is 4 μm, the line spacing is 7 μm, the line width of the BOTTOM layer is 6 μm, and the line spacing is 5 μm.

We connected the two coupled lines according to Figure 1 to obtain a balun graph, as shown in Figure 10b. Port 1 is an unbalanced port; port 2 and port 3 are balanced ports. To simulate the amplitude and phase characteristics of the balun, it should be noted that during the simulation, the balanced port should be terminated with a diode ring, and the unbalanced port should be connected to a 50-ohm load.

Figure 11 shows the simulation results of balun amplitude and phase imbalance. It can be seen that the amplitude and phase imbalance begin to deteriorate at high frequencies. This is because the impedance of the diode ring gradually decreases at high frequencies, and its matching with the balun becomes worse.

#### 3.1.2. The Design of the Mixer

The diode is the core unit to realize the frequency mixing. This paper adopted Schottky diode to realize it. The equivalent circuit of the Schottky diode is shown in Figure 12. 

$R_{s}$ is the parasitic resistance, $R_{j}$ is the intrinsic junction resistance, and $C_{j}$ is the intrinsic junction capacitance.

Compared with the PN junction diode, the Schottky diode has a lower junction resistance ($R_{j}$), and its conduction voltage is lower, so the driving power requirement for the LO signal is lower, and the loss is smaller; the intrinsic junction capacitance ($C_{j}$) of the Schottky diode affects its operating frequency; the smaller the size, the higher the operating frequency, but the greater the loss, the worse the linearity. Therefore, it is necessary to consider the size of the diode comprehensively. The final selected diode had a finger number of 2 and a gate width of 40 μm.

According to the schematic diagram in Section 2.2, connect two baluns and four diodes, add an IF tap on the RF port, and add three RF port pads to form a prototype of the mixer chip, which is also a standard Marchand balun-based mixer. In order to reduce the mismatch at the high-frequency end and expand the bandwidth to higher frequency, a capacitor can be connected in series with the open port of the balun to provide an RF bypass, thereby improving the high-end characteristics. 

The load of the balun in the double-balanced mixer is a quad-ring diode. In the high-frequency band, the loaded capacitor can match the impedance of the balun with the impedance of the quad-ring diode, resulting in a better amplitude and phase imbalance in the high-frequency band. Figure 13 compares the effects of different sizes of capacitors on the high-frequency amplitude and phase imbalance when the balun’s load is a quad-ring diode. It can be seen from Figure 13 that increasing the capacitor can optimize the amplitude and phase imbalance of the high-frequency band, and the size of the capacitor is frequency-dependent. After comparison, a 1.38 pF capacitor was selected for loading.

The improvement in the amplitude and phase imbalance in the high-frequency band represents the expansion of the mixer bandwidth towards high frequencies. The bandwidth evaluation of the mixer can be obtained by checking the conversion loss, fixing the IF frequency, and scanning the RF frequency. Figure 14 compares the conversion loss of the mixer before and after loading a small capacitor. Figure 14a is the conversion loss curve before and after loading the capacitor on the LO balun side. It can be seen that it expanded the bandwidth, assuming a 15 dB loss is the acceptable variable conversion loss for the system; the operating frequency range was about 1.05 GHz to 5.05 GHz before loading the capacitor, with a relative bandwidth of about 1.31. After loading the capacitor, the operating frequency range was about 920 MHz to 5.95 GHz, with a relative bandwidth of about 1.46, an increase of about 10%. In order to make the LO port and RF port interchangeable, a symmetrical structure was designed, so a capacitor of the same size was also added to the ground on the RF-side balun. Comparing the conversion loss curve as shown in Figure 14b, it can be seen that the loss in the low-frequency band was reduced after adding the capacitor.

In order to improve the port isolation, on the basis of the standard structure, a small capacitor was added to the IF port to provide an RF bypass, and a small inductor was connected to form an LC circuit. The position and size of the LC are indicated in Figure 6, with a 1.38 pF capacitor and a 0.8 nH inductor loaded. An LC circuit has low-pass characteristics, which can suppress RF leakage and LO leakage, thereby improving the isolation from the RF to IF ports and the LO to IF ports. Figure 15 shows the leakage of the RF signal and the LO signal at the IF port under different RF powers. It can be seen that in the linear stage, the RF leakage signal decreased by about 8 dB after loading the LC circuit. As the RF power increased, the output IF power increased and the RF leakage signal also increased, with the difference between the two becoming smaller and smaller. Due to the load of the LC circuit, the leakage of the LO signal at the IF port was reduced by about 7 dB.

### 3.2. Final Layout

The 1–6 GHz broadband mixer was designed based on a 0.25 μm GaAs process. As shown in Figure 16, a symmetrical design was adopted for the LO balun and RF balun so the RF and LO ports could be interchanged, basically without affecting the overall indicators. All three ports were designed with GSG for easy probe testing. The chip size of the proposed mixer was 1.13 mm × 0.95 mm.

## 4. Experimental Section

### 4.1. Experimental Platform

The mixer measurement was carried out on a probe platform with a calibrated vector network analyzer, as indicated in Figure 17. The RF and LO signals were output from ports 1 and 2 of the PNA, respectively, and input to the mixer to be tested. The output IF signal of the mixer to be tested was input to port 3 of the PNA (the port settings can be adjusted during calibration). 

### 4.2. Experimental Results

The characteristics of the mixer were obtained under certain input conditions, and the mixer exhibited different characteristics under different LO drive powers. Figure 18 shows the simulated and measured conversion loss curves under different LO input powers. It can be seen that in the simulation curve, the LO power of 14 dBm is an inflection point; when the LO power was higher than this power, the conversion loss basically remained unchanged, and the measured curve also shows similar characteristics but the inflection point is not obvious, which was caused by the poor accuracy of the diode model. According to the test results in Figure 18, 13 dBm was selected as the LO driving power, which is also consistent with the design value.

The bandwidth evaluation of the mixer is performed by fixing the output IF frequency, scanning the RF frequency and changing the LO frequency accordingly, and obtaining the conversion loss curve; when the conversion loss falls below the requirement of the system, it represents the bandwidth cut-off point of the mixer. Figure 19 shows that within the RF frequency range of 1 GHz to 6 GHz, the conversion loss was higher than −14.8 dB, which is in good agreement with the simulation curve. The large fluctuations in the simulation curve around 2 GHz are due to the influence of the simulated gold wire model.

Figure 20 depicts the simulation and measured isolation characteristic curves. The consistency between the measured and the simulated isolation was very good, indicating that the accuracy of the small-signal model is very high. It can be seen from the curve that within the designed bandwidth range, the isolation of LO to RF and LO to IF is basically below 30 dB. Because the IF port was led out from the RF port, the isolation of IF to RF was poor. By cascading an LC circuit to the IF port, the RF signal was suppressed to a certain extent, and the isolation was essentially below 10 dB.

The linearity characteristics of the chip were characterized by testing its input 1 dB compression point. The test conditions here were a fixed LO power of 13 dBm, 2 GHz RF, and 100 MHz IF. The measurement results are shown in Table 1, and the IP1 dB of the chip was about 13.5 dBm.

Linearity and dynamic range are two important indicators in communication systems, so in order to achieve broadband and miniaturization requirements, the bandwidth, input power, isolation, and size of the mixer are the main parameters to evaluate them. The feasibility of a mixer such as that proposed is commonly established using a pertinent figure of merit (FoM) that calibrates its accomplishment. The FoM is included with relative bandwidth (RBW); isolation (ISO), including RF-to-LO isolation and LO-to-IF isolation; input power at 1 dB compression point (IP1dB); conversion loss (CL) and area to meet broadband; high isolation; and a high-linearity system target, all given as Equation (16) [12]: (16)$$FoM=\frac{RBW\times \frac{\mathrm{ISO}\left[dB\right]}{100dB}\times \frac{IP1\mathrm{dB}\left[dBm\right]}{10dBm}}{\frac{\mathrm{CL}\left[dB\right]}{10dB}\times \frac{\mathrm{Area}\left[{mm}^{2}\right]}{1{mm}^{2}}}$$
$$RBW=\frac{f_{max}-f_{min}}{f_{center}}\times 100\%$$

Here, ISO represents the sum of RF-to-LO isolation and LO-to-IF isolation, $f_{max}$ is the maximum operating frequency of the mixer, $f_{min}$ is the minimum operating frequency of the mixer, and $f_{center}$ is the center operating frequency of the mixer.

The larger the FoM value, the better the overall performance. Table 2 encapsulates the measured performance of the proposed mixer and compares it with some passive broadband double-balanced mixers that have similar applications. It is clear from Table 2 that the proposed mixer has the highest FoM value compared to other mixers, indicating better overall performance.

## 5. Conclusions

A 1–6 GHz broadband double-balanced mixer was implemented based on the GaAs pHEMT process. It features wide bandwidth, high linearity, high isolation, and low loss in a small area. Based on the standard Marchand balun mixer, this brief made two improvements and achieved good results. Firstly, through a loading capacitor, the amplitude and phase imbalance of the balun were improved, expanding the bandwidth of the mixer. Secondly, by cascading a first-order LC circuit, the leakage signals of RF and LO were reduced. 

## Figures and Tables

**Figure 1 micromachines-14-02069-f001:**
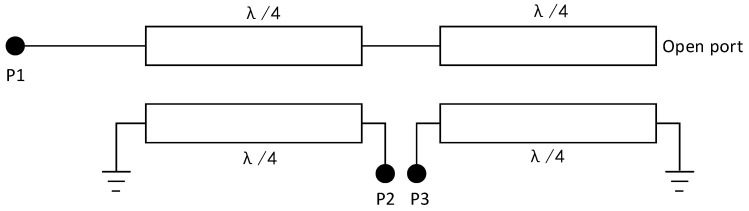
Typical structure of Marchand balun.

**Figure 2 micromachines-14-02069-f002:**
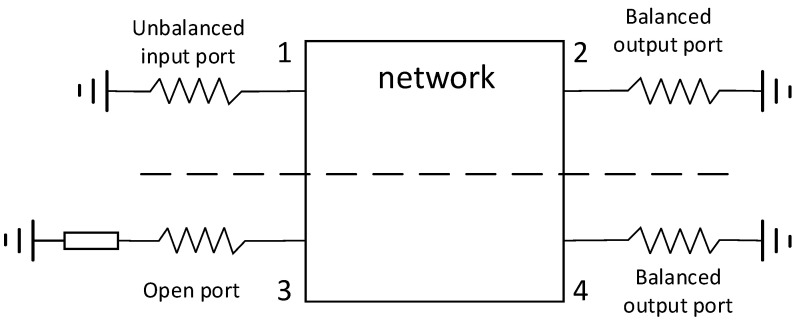
Schematic diagram of four-port network.

**Figure 3 micromachines-14-02069-f003:**
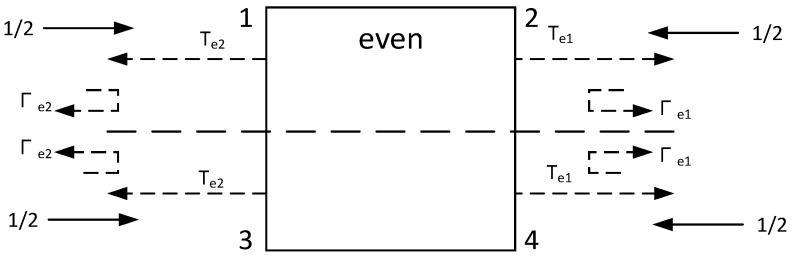
Four-port network under even-mode.

**Figure 4 micromachines-14-02069-f004:**
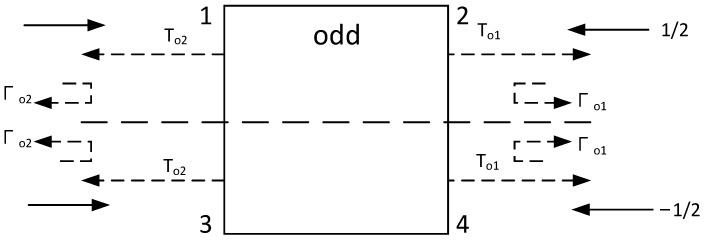
Four-port network in odd mode.

**Figure 5 micromachines-14-02069-f005:**
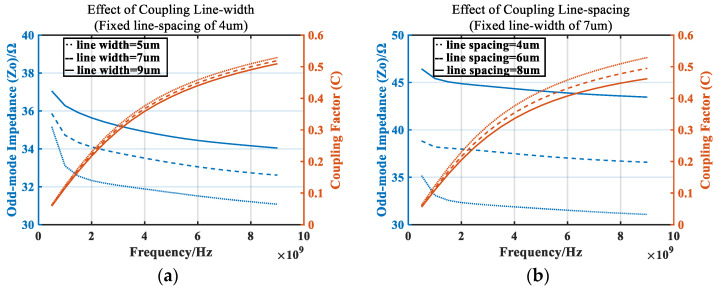
Effect of (**a**) coupling line width and (**b**) coupling line spacing on odd-mode impedance and coupling factor.

**Figure 6 micromachines-14-02069-f006:**
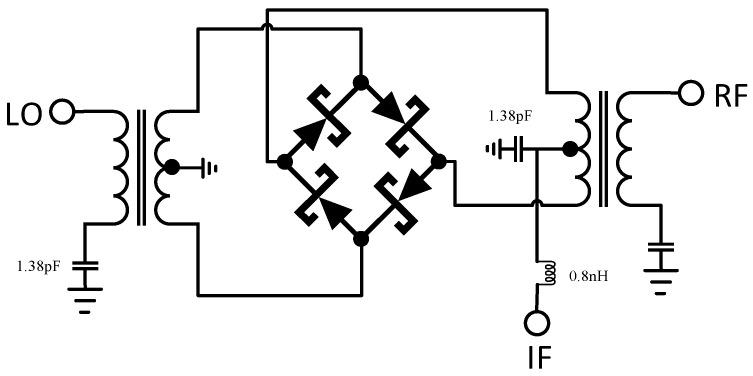
Schematic of a passive double-balanced mixer.

**Figure 7 micromachines-14-02069-f007:**
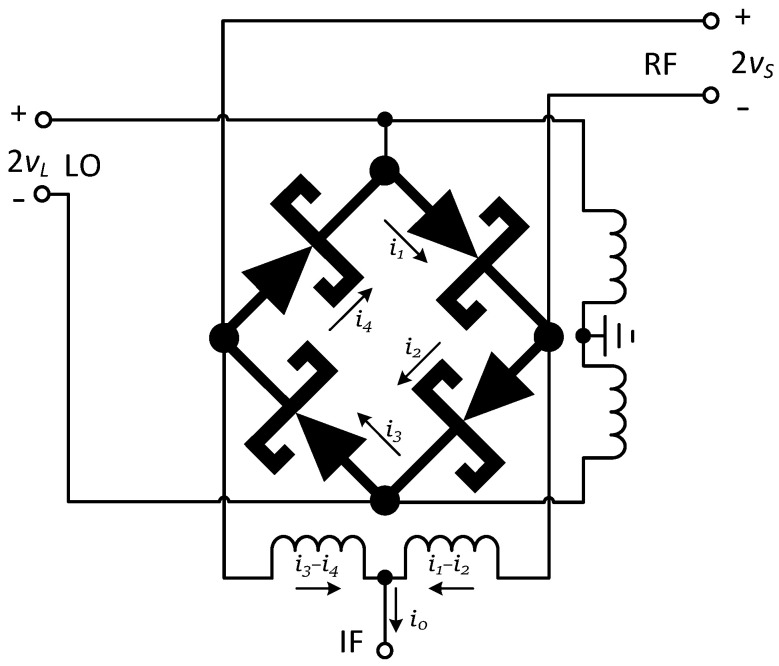
Diode mixing principle.

**Figure 8 micromachines-14-02069-f008:**
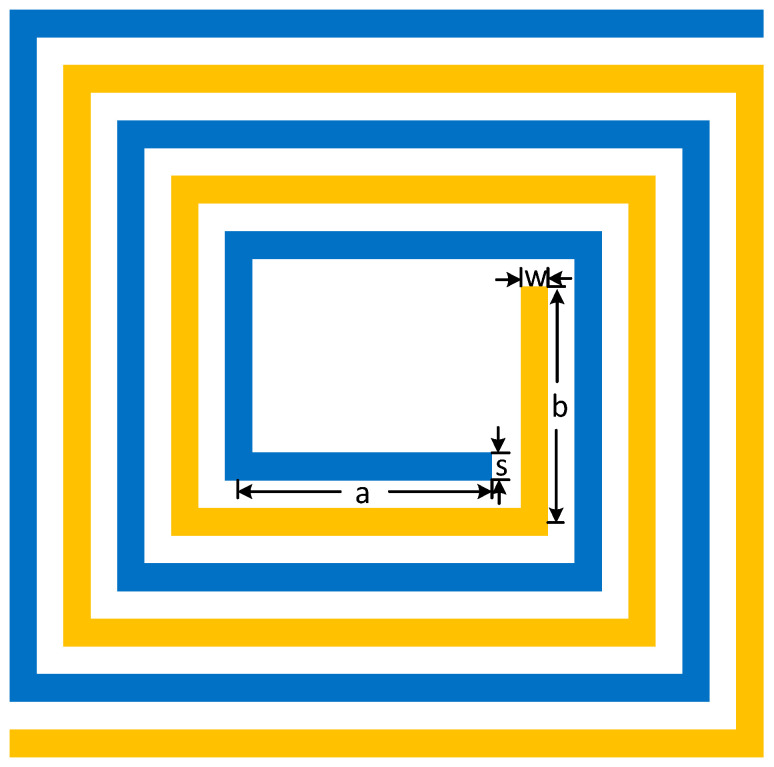
Double-wire winding structure.

**Figure 9 micromachines-14-02069-f009:**
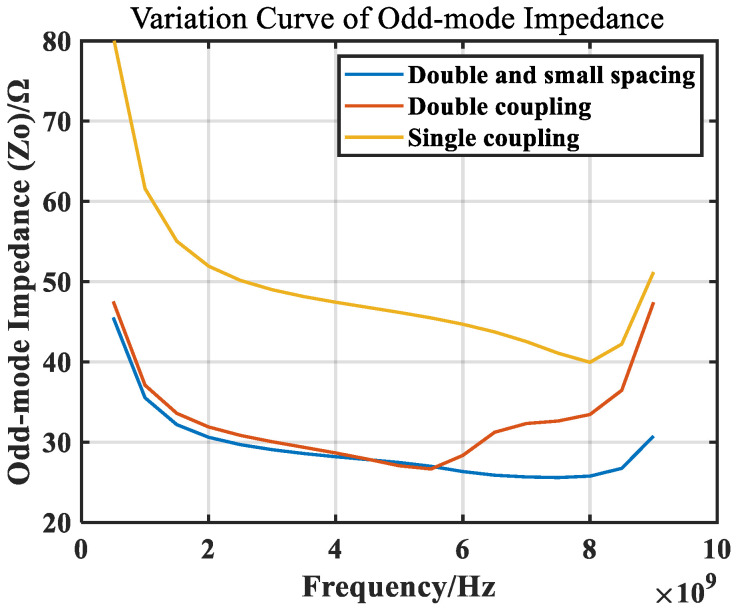
Simulated odd-mode impedance of coupled lines.

**Figure 10 micromachines-14-02069-f010:**
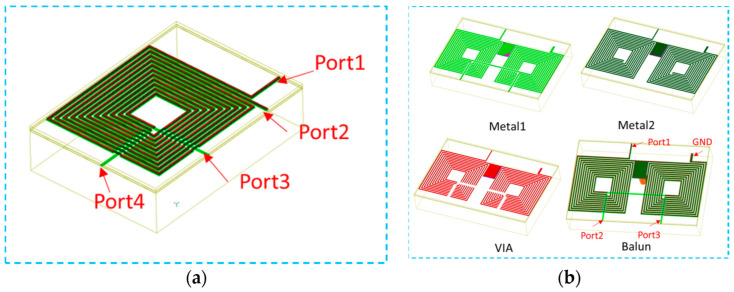
Layout of (**a**) coupled lines and (**b**) balun.

**Figure 11 micromachines-14-02069-f011:**
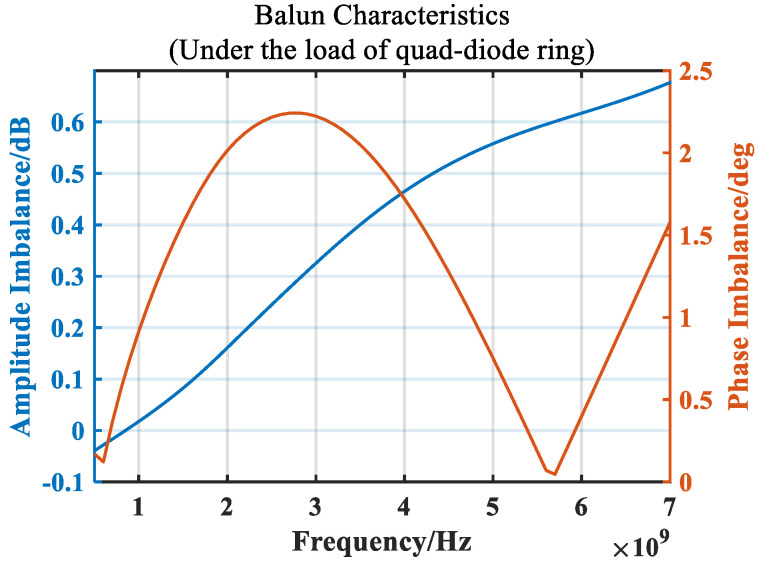
Simulated balun amplitude and phase imbalance.

**Figure 12 micromachines-14-02069-f012:**
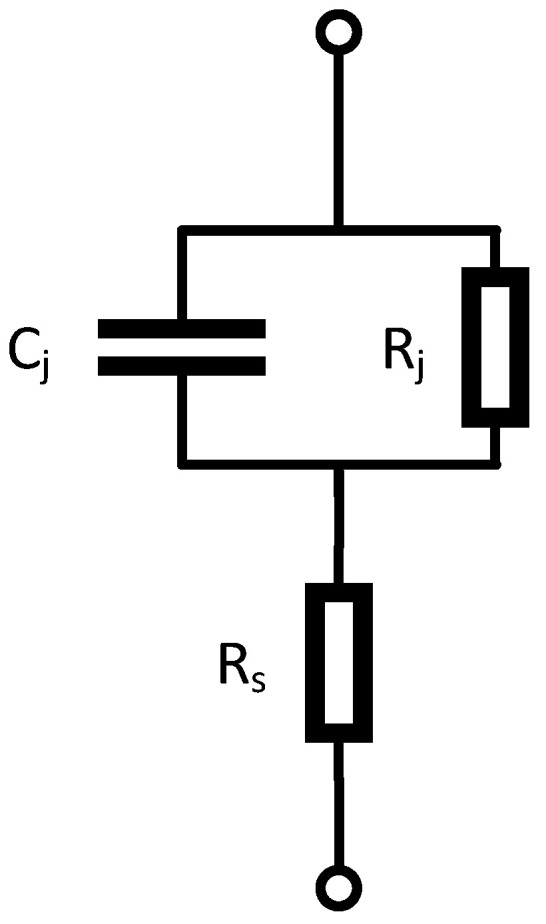
Equivalent circuit AC model of Schottky diode.

**Figure 13 micromachines-14-02069-f013:**
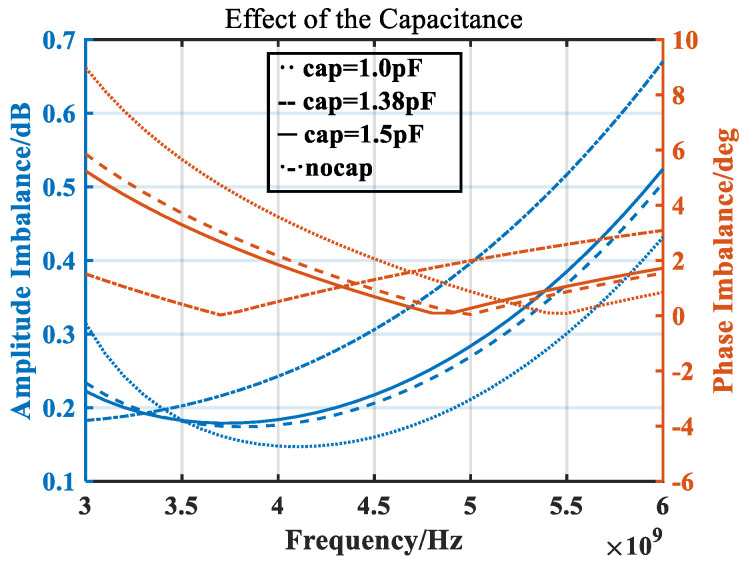
Effect of capacitance on the balun amplitude and phase imbalance.

**Figure 14 micromachines-14-02069-f014:**
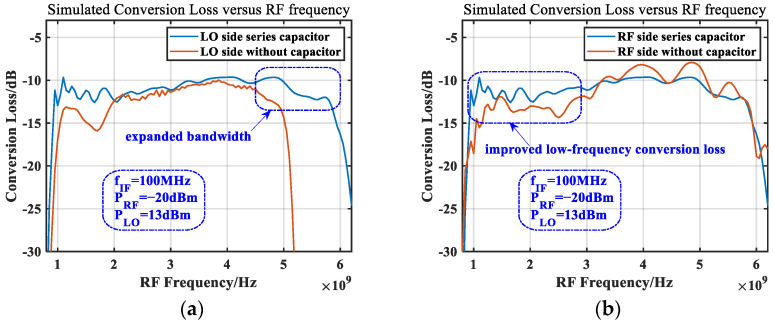
Effect of capacitance to ground on bandwidth. (**a**) LO side and (**b**) RF side.

**Figure 15 micromachines-14-02069-f015:**
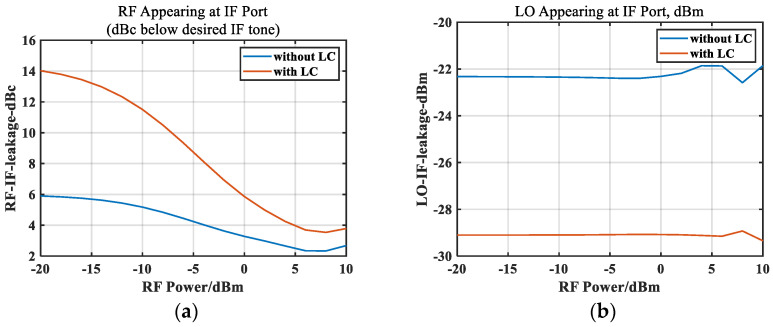
Simulated leakage vs. RF power. (**a**) RF leakage at IF port and (**b**) LO leakage at IF port.

**Figure 16 micromachines-14-02069-f016:**
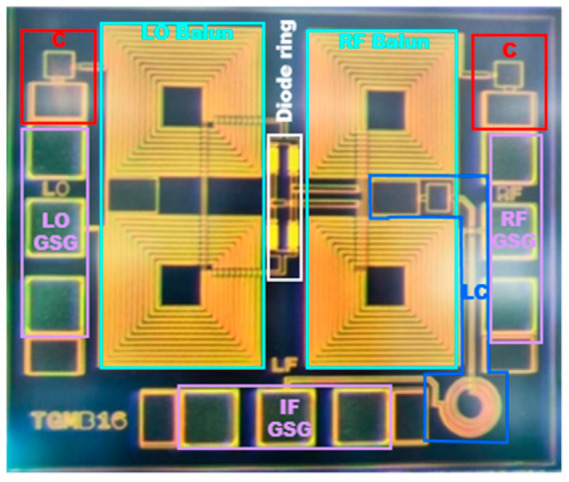
Die photograph of the proposed mixer.

**Figure 17 micromachines-14-02069-f017:**
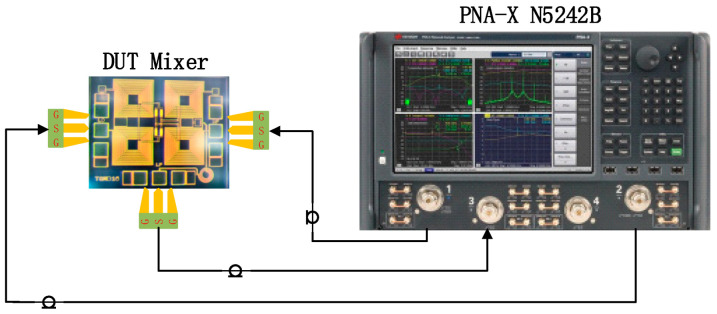
Experimental configuration of the proposed mixer.

**Figure 18 micromachines-14-02069-f018:**
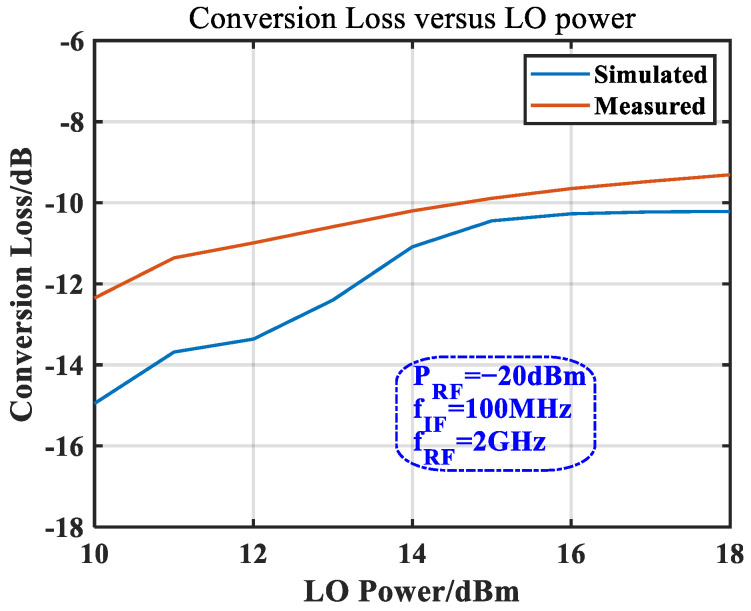
Simulated and measured conversion losses vs. LO power.

**Figure 19 micromachines-14-02069-f019:**
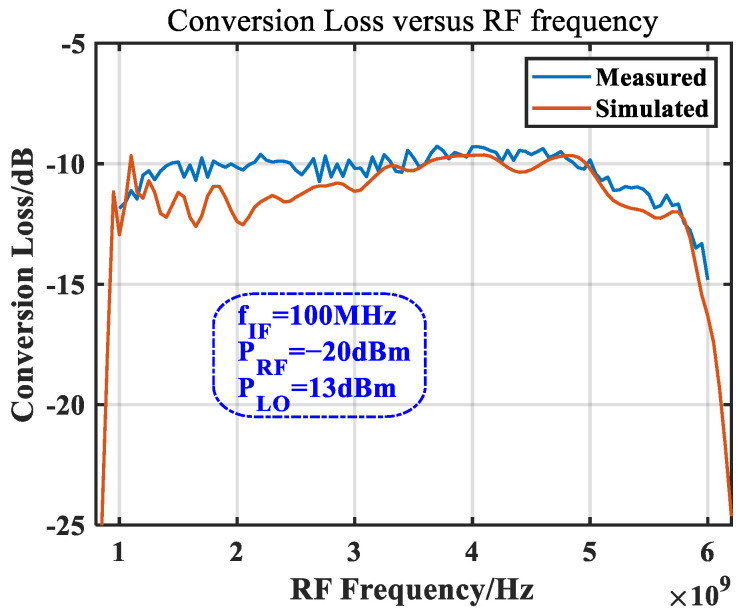
Simulated and measured conversion losses vs. RF frequency at a fixed LO power.

**Figure 20 micromachines-14-02069-f020:**
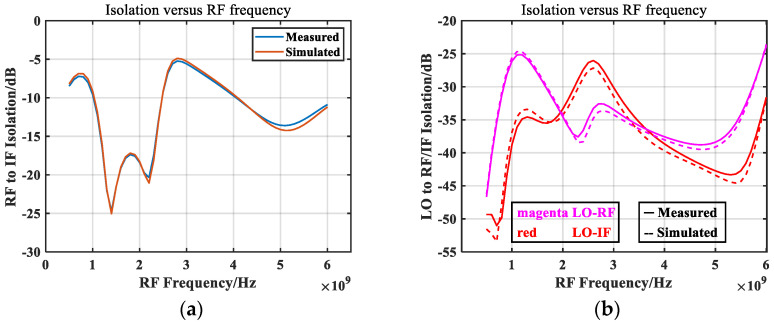
Simulated and measured isolation vs. RF frequency. (**a**) RF to IF and (**b**) LO to RF/IF.

**Table 1 micromachines-14-02069-t001:** Measured compression performance.

RF Input Power (dBm)	IF Output Power (dBm)	Conversion Loss (dB)
−20.0	−30.43	−10.43 (linear)
13.54	2.1	−11.44 (at_P1dB)

**Table 2 micromachines-14-02069-t002:** Double-balanced mixer performance comparison.

	[13]	[7]	[14]	[4]	[3]	[2]	This Work *
Year	2009	2011	2016	2019	2021	2022	2023
Tech.	Marchand single ring	Marchand single ring	Marchand single ring	Marchand single ring	Novel Marchand ring	Marchand double ring	Marchand single ring
Frequency (GHz)	8~20	11~40	1.5~3	2.8~6	2~22	1.5~5	1~6
RBW	0.86	1.14	0.67	0.73	1.67	1.08	1.43
IL (dB)	5~11	11	10.4	8	11.5	7	10.1
IP1dB (dBm)	0~4	12	16	10	11	10	13.5
RF-to-LO isolation (dB)	35	26.9	40.9	40	37	40	34
LO-to-IF isolation (dB)	32	43.2	35	45	20	42	34
Chip area (mm^2^)	1.7 × 1.8	0.85 × 0.85	≈1	1.4 × 1.1	1.7 × 1	1.6 × 1.6	1.13 × 0.95
FoM	0.15	1.20	0.77	0.50	0.53	0.49	1.21

* Test conditions: LO power, 13 dBm; RF frequency, 2 GHz; IF frequency, 100 MHz.

## Data Availability

The data presented in this study are available on request from the corresponding author. The data are not publicly available due to commercial restrictions.

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
