# Peer review of "The Design of a 1–6 GHz Broadband Double-Balanced Mixer"

_micromachines, 2023, doi:10.3390/mi14112069_

Round 1

Reviewer 1 Report

Comments and Suggestions for Authors

The paper gives a broadband mixer MMIC design, and theoretical analysis and design considerations were discussed, as well as test results.

1) The paper analyzed Marchand Balun and presented the relationship between coupling line width, coupling line spacing, and odd-mode impedance etc (Figure 5). However, the values of the coupling line width and coupling line spacing in Figure 5 are relatively large, which differs greatly from the actual dimensions used in the circuit (analytical coupling line spacing of 12.7 μm~25.4 μm v.s. actual line spacing of 7 μm) . Please use odd and even mode analysis based on the actual dimensions in the MMIC chip to better demonstrate the actual work of the paper.

2) The paper says "by connecting a capacitor in series with the open port of the balun to the ground, the relative bandwidth of the mixer is increased by 0.1." The added capacitor needs to be represented in the circuit diagram of Figure 6, and what is the capacitance value? How is it determined specifically?

Also, how to determine the relative bandwidth of the mixer is increased of 0.1? Please provide detailed information on the bandwidth before and after adding capacitor.

3) The paper states that an LC filter is used at the IF port to improve the isolation of RF-IF. This also needs to  be represented in the circuit diagram of Figure 6, and what is the device value of the LC used? And how to determine the value of the devices?

4) The Balun structure in Figure 10 is not clear. It is recommended to represent in the form of separated layers, stacked 3D, etc. Meanwhile, are the physical dimensions of the RF and LO balun implemented the same? And what is the insertion loss achieved?

5) The English writing needs to be strengthened and improved

Comments on the Quality of English Language

Please carefully check multiple places in the paper, such as "high end bandwidth” etc

Author Response

Dear Editor and Reviewer

Thank you for your valuable feedback. We have made revisions to the manuscript. Please see the attachment for details.

Reviewer 2 Report

Comments and Suggestions for Authors

This work is timely and of utmost importance. However, the purpose of the work is not clear. It is good to provide a section on how this is going to affect current and future technology. Further, in the current version, it seems the novelty of the work is scant. It would be great to put a section about the novelty of the work. 

It would be great to specify the novelty of the work in a separate paragraph in the introduction.

In the abstract, "First, by connecting a capacitor in series with the open port of the balun to the ground, the relative bandwidth of the mixer is increased by 0.1." this statement is not fully clear.

I suggest including the reasoning: "The high bandwidth and small size of double-balanced mixers often cannot be satisfied at the same time."

Also, Include the requirement/application of such a system.  

Added, I suggest including the pros and cons of going higher in the frequency range, as the research domain is more interested in terahertz frequencies.

It is good to separate Figure 14 into separate figures for better understanding.

Why is the increased BW effect not observed in Figure 13 b? Also, please provide the amount of average improvement with and without series capacitors in the dB scale in writing to show the benefit of the system.

It would be better to increase the size of Figure 8 and explain it more with reference.

It is better to improve the positioning of Table II. 

Please increase the size of the Figures; Important messages are sometimes not visible. 

There are several typos; please consider correcting them. Such as - please use In [1], designed rather than [1] designed, and many others.

Comments on the Quality of English Language

There are several typos and grammatical mistakes that need to be addressed. Further, on several occasions, it feels that the sentence is not clear and complete, such as "Compared to standard Marchand balun mixers, this design has two techniques" and many more. Please address them. 

Author Response

(The authors gave the same response as above.)

Round 2

Reviewer 1 Report

Comments and Suggestions for Authors

The revised manuscript has improved some unclear areas in the paper, and it is also necessary to explain or analyze the reasons for the significant curve jitter of the simulation curve in Figure 19, especially at low frequencies.

Comments on the Quality of English Language

The paper has reasonable readability

Author Response

Dear Editor and Reviewer

Thank you for your valuable feedback. Attached is the response to the feedback. Please check it.

Reviewer 2 Report

Comments and Suggestions for Authors

Thank you for the changes made by the authors and for answering the comments in rebuttal. The paper depicts the thoughts and design details clearly after revision. 

Comments on the Quality of English Language

N/A

Author Response

Dear Editor and Reviewer

Thank you for reviewing the manuscript again. We are very pleased to see your approval of the manuscript. Thank you again.